# Transcriptome Profiling, Biochemical and Physiological Analyses Provide New Insights towards Drought Tolerance in *Nicotiana tabacum* L.

**DOI:** 10.3390/genes10121041

**Published:** 2019-12-15

**Authors:** Rayyan Khan, Peilu Zhou, Xinghua Ma, Lei Zhou, Yuanhua Wu, Zia Ullah, Shusheng Wang

**Affiliations:** 1Tobacco Research Institute, Chinese Academy of Agricultural Sciences, Key Laboratory of Tobacco Biology and Processing, Ministry of Agriculture, Qingdao 266101, China; rayyanswb@gmail.com (R.K.); 82101172159@caas.cn (L.Z.); wuyuanhua@caas.cn (Y.W.); zianust512@gmail.com (Z.U.); wangshusheng@caas.cn (S.W.); 2College of Agronomy, Resource and Environment, Tianjin Agricultural University, Tianjin 300384, China; zhpeilu@163.com

**Keywords:** *Nicotiana tabacum*, drought stress, tolerance, transcriptome, RNA-Seq, plant hormones, proline, antioxidant enzymes, varieties

## Abstract

Drought stress is one of the main factors limiting crop production, which provokes a number of changes in plants at physiological, anatomical, biochemical and molecular level. To unravel the various mechanisms underpinning tobacco (*Nicotiana tabacum* L.) drought stress tolerance, we conducted a comprehensive physiological, anatomical, biochemical and transcriptome analyses of three tobacco cultivars (i.e., HongHuaDaJinYuan (H), NC55 (N) and Yun Yan-100 (Y)) seedlings that had been exposed to drought stress. As a result, H maintained higher growth in term of less reduction in plant fresh weight, dry weight and chlorophyll content as compared with N and Y. Anatomical studies unveiled that drought stress had little effect on H by maintaining proper leaf anatomy while there were significant changes in the leaf anatomy of N and Y. Similarly, H among the three varieties was the least affected variety under drought stress, with more proline content accumulation and a powerful antioxidant defense system, which mitigates the negative impacts of reactive oxygen species. The transcriptomic analysis showed that the differential genes expression between HongHuaDaJinYuan, NC55 and Yun Yan-100 were enriched in the functions of plant hormone signal transduction, starch and sucrose metabolism, and arginine and proline metabolism. Compared to N and Y, the differentially expressed genes of H displayed enhanced expression in the corresponding pathways under drought stress. Together, our findings offer insights that H was more tolerant than the other two varieties, as evidenced at physiological, biochemical, anatomical and molecular level. These findings can help us to enhance our understanding of the molecular mechanisms through the networks of various metabolic pathways mediating drought stress adaptation in tobacco.

## 1. Introduction

Drought is one of the most eminent environmental stresses and defined as a deficit of water relative to normal conditions [1] or a sustained period of below-normal water availability [2]. Drought stress is a serious threat to agriculture and food security and it has been exacerbated due to the drastic and rapid changes in global climate [3]. It affects the production of both rain-fed and irrigated crops. As plants are sessile in nature, they respond to drought stress in a series of processes at physiological, biochemical and molecular level, allowing them to escape or adapt to unfavorable conditions [4,5]. Furthermore, drought impairs nutrient uptake and transport, inhibits photosynthesis, loss of cell turgidity and accumulation of reactive oxygen species, crop growth and productivity and in extreme cases crop failure [6,7]. 

Tobacco (*Nicotiana tabacum* L.) is a broad nonfood grown crop of Solanaceae: an economically important family of flowering plants. It is also used as a model plant for physiological and genetic studies [8]. Tobacco is an economic crop worldwide, its leaves are the primary product and its productivity is vulnerable to drought. The production loss of the crop depends upon the intensity and duration of drought and also on the developmental stage of the plants at which drought stress occur [9]. Throughout the entire life cycle of flue-cured tobacco, drought stress at various stages, such as the rejuvenation stage, group stage, vigorous growth stage, mature stage etc. (from vegetative to flowering) affect the growth, yield and quality [10,11].

An unavoidable consequence of drought is the enhanced production of reactive oxygen species (ROS) in the various cellular compartments. ROS cause cell damage and death by engulfing the scavenging ability of the plant defense system [12]. Due to water stress, plants adapt themselves through various defense mechanisms and maintain cellular redox homeostasis to counteract the ROS by various antioxidant substances and enzymes [13]. When plants face unfavorable conditions, it perceives stress signals from the surroundings; various physiological, biochemical and molecular modifications occur. Some of these changes just represent a consequence of the cell damage, while other show adaptive processes that plants evolved in response to the stressful environmental cues. At a molecular level, plants respond and adapt to drought by inducing both regulatory and functional genes via perception, signal transduction, expression of downstream drought responsive genes network and production of metabolites that protect and maintain the structure of cellular components [14].

Drought responses are notoriously multigenic and quantitative with strong environmental effects on phenotypes. Various studies found that plants bring adaptation to drought by the participation of multiple pathways in response to drought stress, i.e., phenylpropanoid biosynthesis and peroxisome pathway [15], metabolic pathway and biosynthesis of secondary metabolism [16], hormone-mediate signaling and biosynthesis of lignin [17]. A delicate balance between the multiple pathways is necessary for plant growth and development, which might be broken when plants face unfavorable conditions. Abscisic acid (ABA) is an important phytohormone and play crucial roles in regulating the abiotic stress responses [18]. The various regulatory pathways crosstalk with each other to form a drought-defensive network as the antagonistic actions of various components of the ABA and cytokinin signaling pathways to mediate drought stress responses and pin-point how plants coordinate growth and drought stress responses by integrating multiple hormones pathways [19].

Drought has become a very important limiting factor in the global production of tobacco, as it requires plentiful amount of water throughout its life cycle. However, presently, drought stress has become a limiting factor for its growth and development. In recent years, different approaches, such as physiological, transcriptomics and metabolomics, have been used in various studies, which highlighted the importance of disease resistance, abiotic stresses tolerance and drought tolerance in both transgenic and non-transgenic tobacco plants [20,21,22,23,24]. Similarly, other studies also put light on the role of various pathways such as plant hormones signal transduction pathway, proline metabolism pathway, starch and sucrose metabolism pathway etc. in response to drought stress [25,26,27,28]. The complete and comprehensive physiological, anatomical, biochemical and transcriptomic analyses are rare under drought stress in flue-cured tobacco cultivars. Therefore, it is better to know that how different tobacco varieties respond to drought stress at multiple levels in plants in order to improve drought tolerance. In the present work, the transcriptomic analysis using RNA sequencing was performed on the leaves of the three tobacco cultivars, i.e., NC55 (N), HongHuaDaJinYuan (H) and Yun Yan-100 (Y) plants under drought stress in order to investigate the pathways involved in responses to drought stress and explore the drought tolerance mechanism and other inputs will serve as useful resources for establishing functional role of the molecular events during drought stress responses in tobacco. 

## 2. Materials and Methods

### 2.1. Plant Materials and Drought Stress Treatment

*Nicotiana tabacum* cvs. NC55, HongHuaDaJinYuan and Yun Yan-100 were used in this study. After germination, the tobacco seedlings were grown in the floating breeding system in a growth room. The temperature was approximately 26 °C with 45% relative humidity, a photoperiod of 12/12-h (day/night) and a light intensity of 300 µmol m^−2^ s^−1^. Only plants with similar growth and development were used in the experiment. Drought stress was given to tobacco seedlings by removing water from the floating breeding system. Samples from control (CK) and drought-treated plants (DS) at different time intervals were collected and frozen immediately in liquid nitrogen and kept at −80 °C until further use. The sketch of the experiment and plant treatments is portrayed in Figure 1. 

### 2.2. Determination of Fresh and Dry Biomass, Chlorophyll Content 

Plants fresh weight was recorded by separating plants into shoot and roots. After fresh weight, all the plant parts were dried (80 °C) until constant dry weight was achieved and weighed.

Total chlorophyll content was determined as described by [29]. Briefly, the fresh leaves were discolored in 80% acetone. The absorbance was measured at 663 nm and 645 nm using a spectrophotometer.

### 2.3. Chlorophyll Fluorescence and Multicolor Fluorescence Imaging 

The leaf chlorophyll fluorescence was measured on the second fully expanded leaf from the apex in dark- and light-adopted leaves using a PAM (Pulse amplitude modulated) fluorescence imaging system (FluorCam FC 800, Photon Systems Instruments, Brno, Czechia). The protocol of [30] was essentially followed to obtain the kinetic chlorophyll fluorescence parameters and images at various time intervals upon the onset of drought stress. Based upon the basic chlorophyll signals the maximal photochemical efficiency (Fv/Fm), steady-state PSII quantum yield (ΦPSII_Lss) and steady-state non-photochemical quenching (NPQ_Lss) were calculated. 

The multicolor fluorescence imaging was carried out by the excitation of leaves using a UV (ultraviolet) (320–400 nm) LED panel as an excitation source. The fluorescence images were acquired in blue (BF-440 nm), green (GF-520 nm), red (RF-680 nm) and far-red (IrF-740 nm) regions.

### 2.4. Determination of Leaf Water Potential

Leaf water potential (LWP) was measured as described previously [31]. Leaves of the same size and position from the three tobacco varieties of control and drought-stressed plants were sampled. Disks from the leaves were sealed in C-52 psychrometer chamber connected to the PSYPRO water potential system (WESCOR, Logan, UT, USA). The values of Ψ_leaf_ were measured as MPa.

### 2.5. Measurements of Superoxide dismutase (SOD), Peroxidase (POD), and Catalase (CAT)

The activities of these enzymes in three tobacco cultivars i.e., N, H and Y, in control and drought-treated plants were measured spectrophotometrically.

The activity of SOD was assayed by measuring its ability to inhibit the photochemical reduction of nitro blue tetrazolium by reading the absorbance of the reaction mixture at 560 nm [32]. 

The activity of POD was monitored at 470 nm absorbance by the oxidation of guaiacol and one unit of the enzyme activity is the amount of enzyme that changes the absorbance by 0.01 per minute [33]. 

CAT activity was determined by the decomposition of H_2_O_2_. The absorbance was recorded at 240 nm [32].

### 2.6. Quantification of Ascorbc acid (AsA), and Glutathione (GSH) Contents

Briefly, leaves were homogenized in 5% phosphoric acid and EDTA (Ethylenediamine tetraacetic acid) and then centrifuged at 11,500 rpm for 15 min at 4 °C. The supernatant was collected for the quantification of ASA and GSG content. ASA content was assayed as the reduction of Fe^3+^ to Fe^2+^ by ascorbic acid in acid solution. The absorbance was recorded as Fe^2+^ formed a chelate with bipyridyl at 525 nm [34]. 

The reduced glutathione (GSH) was determined in control and stressed tobacco seedlings of the various cultivar as the reduction of 5,5′-dithiobis(2-nitrobenzoic acid) (DTNB) was spectrophotometrically measured at 412 nm in the supernatant [34,35]. 

### 2.7. Determination of Proline and Soluble Sugars

Briefly, proline levels were determined as fresh leaves were homogenized in 3% sulfosalicylic acid. The homogenate was centrifuged at 12,000 rpm for 10 min. Then, the supernatant was boiled at 100 °C for 30 min with the addition of glacial acetic acid and ninhydrin solution to it. The absorbance was read at 520 nm after cooling the mixture and addition of methylbenzene [36].

Soluble sugars were quantified with the phenol-sulfuric acid method [37]. Soluble sugars were determined by homogenization of leaf samples with deionized water and then the extract treated with phenol and sulfuric acid. The absorbance of the extract was determined at 485 nm by a spectrophotometer. 

### 2.8. Determination of Malondialdehyde (MDA) and Hydrogen Peroxide (H_2_O_2_)

MDA content as examined by a homogenizing sample of fresh leaves in 5% trichloroacetic acid (TCA) and centrifuged for 15 min at 12,000 rpm. The supernatant was added to a mixture of TCA, butylated hydroxytoluene and thiobarbituric acid and heated for 30 min. The reaction was stopped by cooling the mixture solution in ice. The absorbance of the supernatant was read at 532 and 600 nm [38].

Hydrogen peroxide (H_2_O_2_) content was measured in leaf samples by following the method of [38]. Briefly, the assay mixture having ferrocytochrome c by its oxidation which provides a measure of H_2_O_2_ production. It was determined spectrophotometrically at 550 nm. 

### 2.9. Histological Studies

For anatomical studies, the samples of leaves from the control and drought stressed tobacco plants were taken and fixed in 70% FAA (formalin: acetic acid: alcohol) solution [39]. Plant tissues were then dehydrated through a series of ethanol. Specimens were then embedded in paraffin. The specimens were then sectioned into slices, stained, cleared with xylene and then mounted. They were viewed and photographed on a microscope to detect histological manifestations of noticeable responses resulting from the drought stress.

### 2.10. RNA Extraction, cDNA Library Construction, and Transcriptome Sequencing

RNA samples were prepared from 18 harvests (2 treatments × 3 genotypes × 3 biological replicates) of tobacco plants. Total RNA was extracted using the TRIzol Reagent (Invitrogen, Carlsbad, CA, USA) following the manufacturer instructions. After RNA extraction, its quality and concentration were determined by agarose gel electrophoresis and NanoDrop spectrophotometer. A more accurate RNA quantification was performed by using an Agilent 2100 Bioanalyzer system (Agilent Technologies, Santa Clara, CA, USA). Subsequently, via magnetic beads with Oligo (dT), the mRNA of the samples were enriched. Briefly, a single stranded and double stranded cDNA was synthesized from the mRNA using random hexamers and AMPure XP beads (Beckman Coulter, Beverly, CA, USA), respectively, and finally, PCR enrichment was performed to obtain final cDNA libraries. After the library was constructed, the libraries concentrations were assayed using Qubit 2.0 (Life Technologies, Thermo Fisher Scientific Inc.) and subsequently examined using Agilent 2100. The libraries were further quantified by the quantitative PCR method. Finally, the different libraries were qualified and pooled then sequenced on Illumina HiSeq 2000 using the platform of Novogene Company (Tianjin, China).

### 2.11. RNA Sequence Data Analysis

Clean reads extracted from raw reads using HISAT were compared with the reference genome *Nicotiana tabacum* var. K326 to get mapped reads [40]. Based on the available data, we also performed mapped region distribution and can be classified as exons, introns, and intergenic regions. Gene expression analysis was performed using Cufflinks. The normalized Fragments Per Kilobase of transcripts per Million mapped reads (FPKM) values > 1 were used for the expression levels of genes from RNA-Seq data [41]. The threshold for DEGs was set as adjusted *P*-value was < 0.01 and |log_2_ fold change| ≥ 2 between the drought-stressed and well-watered tobacco plants libraries of the three varieties. 

### 2.12. Validation of RNA-Seq Data through Quantitative Real-Time PCR (qRT-PCR) Analysis

The qRT-PCR was performed using the total RNA from the leaves of the three tobacco varieties seedlings under both control and drought-stressed conditions, which were exact the same samples that were used for RNA-Seq. The qRT-PCR was carried out on total 20 µL reaction volume following the instructions of TB Green^TM^
*Premix Ex Taq*^TM^ II kit (TaKaRa, Shiga, Japan). The *Actin* gene was used a reference gene and the genes-specific primers of the selected DEGs were designed using PRIMER3 (https://www.ncbi.nlm.nih.gov/tools/primer-blast/). The primers details are listed in Appendix A. 

### 2.13. Statistical Analysis

The data were analyzed by Statistix 8.1 and presented as the mean ± SD (Standard deviation) of the three replicates. The differences in the mean values of the three varieties between well-watered and drought treatment plants were analyzed using a LSD (least significant difference) test at *P* < 0.05. The graphs were made using OriginPro 9.1 (OriginLab Corporation, Northampton, MA, USA) and Excel.

## 3. Results

### 3.1. Effect of Drought Stress on Plant Growth, Chlorophyll Content and Leaf Water Potential of Three Tobacco Varieties

To analyze different responses of the three tobacco varieties in response to drought stress at the different time points, plants were divided into two groups, one group of plants was subjected to drought stress and the other group of plants was kept as a control (well-watered). The phenotypes of the three tobacco varieties in response to drought stress at various time points are shown in Appendix A. The plant fresh weight and dry weight of all the three varieties showed significant differences and reduced when subjected to drought stress (Figure 2A,B, respectively). The plant fresh weight of N, H and Y drought-stressed seedlings was reduced by 29%, 18% and 23% and dry weight by 23%, 7% and 10%, respectively, compared with their individual controls. In summary, H was the least drought affected variety compared to the other two varieties, as less significant reduction was witnessed in the plant fresh and dry weights compared with their controls.

The chlorophyll content showed significant changes and a reduction was observed in the three tobacco varieties in response to drought stress after 72 h of its imposition (Figure 2C). The estimation of chlorophyll content unveiled that the reduction percentage in the chlorophyll content from well-watered conditions to drought-stressed conditions in N, H and Y was 21%, 19% and 21%, respectively. Finally, the dynamic changes in the leaf water potential (LWP) of the three tobacco varieties were examined during the progressive drought stress (Figure 2D). There was a significantly pronounced decrease in the LWP after 72 h of drought stress in the three varieties. Drought stress caused 0.79 fold reduction in LWP of Y, while, 0.78 and 0.72 fold reduction in the LWP of N and H, respectively, when compared to the controls. 

### 3.2. Chlorophyll Fluorescence Parameters in Response to Drought Stress

Figure 3A presented the effect of drought stress at various time points on maximal photochemical efficiency (Fv/Fm) in three tobacco varieties. Drought stress significantly affected the Fv/Fm of only Y at the 48 h time point and was decreased in plants under water deficit conditions in relation to the well-watered condition. The Fv/Fm of N, H and Y showed a significant reduction in drought-stressed plants upon 72 h of drought stress. The maximum photochemical efficiency of N, H and Y were decreased by 8%, 13% and 10% by drought stress upon 72 h of drought stress, respectively. As a result, N, H and Y also exhibited variations in steady-state photochemical efficiency (ΦPSII_Lss) and the parameter representing the regulated dissipation of light energy as heat (NPQ_Lss (steady-state non-photochemical quenching)) in response to drought stress imposition at different time duration (Figure 3B,C, respectively). Only N and Y showed a significantly negative response of ΦPSII_Lss to drought stress at both 48 and 72 h time points. The actual photochemical efficiency of H showed a 13% reduction after 72 h of drought stress, as its value dropped from 0.43 to 0.38, compared with the control. Similarly, 8% and 13% reduction in the ΦPSII_Lss of N and 8% and 17% reduction in Y was witnessed in water-stressed plants upon 48 and 72 h of drought stress, respectively. It was evidenced that water stress had also a significant effect on the NPQ_Lss in H and its value increased from 0.81 in control to 0.97 with a 19% increment upon 72 h of drought stress. The NPQ_Lss of N and Y was also found significant upon 48 and 72 h of drought stress. 5% and 24% increment in the NPQ_Lss of N and 7% and 40% increment for Y were witnessed in drought-stressed plants, compared to the controls, upon 48 and 72 h of drought stress respectively.

Figure 4 presents the representative chlorophyll fluorescence images, including NPQ_Lss (A,B) and ΦPSII_Lss (C,D). The intensity of NPQ_Lss of Yun Yan-100 in the drought stress treatment increased compared with the control, and it spreads from leaf tip and margins to entire tested leaf surface, while H and N had no variation in the intensity of NPQ_Lss. Similarly, images of ΦPSII_Lss of H and N showed no variation, while Y exhibited a decrease in its intensity compared with the control; this trend spread from the leaf tip and edges to the entire tested leaf indicating the spatial heterogeneity. 

Basic multicolor fluorescence parameters, including BF, GF, RF, IrF and one ratio, i.e., F690/F740, which is related to the chlorophyll content, were selected from the full data set (Appendix A). The plant emits blue and green fluorescence from phenolic compounds and red and far-red fluorescence from chlorophyll upon UV light excitation [42]. The BF, GF, RF and IrF values of the three varieties showed significant differences in response to drought stress. The BF, GF and RF values were increased in drought-stressed plants at the 48 and 72 h time points of drought stress, while the IrF values were reduced in drought-treated plants compared with that of the control. Similarly, a significant rise was observed in F690/F740 ratio in drought-stressed plants at 48 and 72 h of drought stress in the three varieties except for H at 48 h, compared with the control. The representative images of BF and GF of the three varieties under drought stress and normal conditions at various time points are shown in Figure 4E. The increase of the BF after drought stress appears more prominently at the edges of the leaves at 48 and 72 h. Similarly, the GF was also increased and more conspicuous upon 48 and 72 h of drought stress. It was spread all over the leaf surface and more obvious at the leaf tip and margins causing spatial heterogeneity within the leaf. 

### 3.3. Oxidative Damage (MDA and H_2_O_2_ Contents) in Response to Drought Stress

Drought stress induced a larger production of malondialdehyde (MDA), a product of lipid peroxidation and H_2_O_2_ in the leaves of N, H and Y seedlings as shown in Figure 5A,B, respectively. After 48 h of drought stress, the MDA contents were increased by 11% in H and 10% in Y. After 72 h of drought stress, the MDA contents was more enhanced by 19%, 14% and 31% in N, H and Y, respectively, in drought-stressed plants compared to well-watered plants. The MDA content in Yun Yan-100 was dynamically increased in drought-treated plants from 10% to 31% as drought stress passed from 48 to 72 h as compared with N and H. Likewise, the dynamic changes in the H_2_O_2_ accumulation levels were also examined in N, H and Y seedlings during 72 h of drought stress. 31% and 18% increment in H_2_O_2_ contents was measured in drought-treated seedlings of Y at 48 and 72 h of drought stress, respectively, in relation to its control. Similarly, 30% and 12% increment in H_2_O_2_ contents were measured in water-stressed plants after 72 h drought stress imposition of N and H, respectively, compared with the controls. 

The oxidative damage in term of MDA and H_2_O_2_ contents was more noticeable in Yun Yan-100 and NC55 as compared to HongHuaDaJinYuan in response to drought stress. 

### 3.4. Antioxidant Enzymes Activity in Response to Drought Stress

The progressive changes in the antioxidant enzymes (SOD, POD and CAT) were quantified in the three tobacco varieties in response to drought stress. The activity of enzymes involved in ROS scavenging, i.e., SOD, POD and CAT, increased significantly in the three tobacco cultivars upon drought stress imposition at various time intervals (Figure 6A–C). SOD for Y was found to be non-significant during the course of the experiment. The activity of SOD was increased from 15% to 55% for N and from 7% to 11% for H as drought stress proceeded from 48 h to 72 h, respectively. The activity of POD in cultivar’s N, H and Y at drought stress for 24 h and only for Y at 48 h of water stress was not significantly affected, while its activity for the cultivars N and H at 48 h of water stress and all the three cultivars at 72 h of drought were significantly affected (Figure 6B). There was a gradual increase in the activity of POD in NC55 from 14% to 21%, and in H from 80% and 60% in drought-treated seedlings, compared with the controls, as drought stress continued from 48 to 72 h. The activity of POD was significant only upon 72 h duration of drought stress in Y and its activity percentage was 57% more in the drought-treated plants related to its control. Similarly, when plants were exposed to drought stress for a different time of duration, the NC55, HongHuaDaJinYuan and Yun Yan-100 behave differently in terms of CAT activity (Figure 6C). The CAT activity was more definitive in N and H drought-treated plants when subjected to 48 h duration and its activity was increased up to 48% and 43%, correlated to the controls, respectively, while in Y, its activity was improved by 25% only. As drought prolonged to 72 h, CAT activity was increased in order like H (41%) > N (31%) in drought-treated plants with respect to the controls. 

In summary, the enhanced activity of various antioxidant enzymes in the ROS system could occur in almost in all kind of adverse environmental conditions. Here, in this scenario, the activity of the enzymatic component of the antioxidants defense system was improved in response to drought stress in a varied way to different varieties. Overall, the performance of HongHuaDaJinYuan was better in term of the percentage increment in POD and CAT enzymes activity in drought-stressed plants in comparison with control; however, NC55 had a better SOD activity than the other two varieties and also a higher POD absolute value. Therefore, HongHuaDaJinYuan and NC55 showed better antioxidant enzymatic performances than Yun Yan-100 in response to drought stress.

### 3.5. Non-Enzymatic Antioxidant Components in Response to Drought Stress

The content of ascorbic acid (AsA) and GSH was quantified in leaves of the three cultivars in order to evaluate whether AsA and GSH-mitigation of oxidative damages in plants due to drought stress was associated with changes in the non-enzymatic antioxidants production (Figure 6D,E, respectively). Imposition of drought stress at different time duration’s significantly affected the content of AsA in the three cultivars. The AsA content was significantly increased for H (13% and 7%) and Y (17% and 13%) as drought stress prolonged from 48 to 72 h in drought-treated plants, respectively, as compared with the controls. Likewise, 19% increment was observed in AsA content for NC55 drought-treated plants at 72 h of drought stress when compared with the control. Similarly, drought stress also caused alterations in the content of GSH in the plants of N, H and Y when subjected to drought stress. Drought treatments for 48 and 72 h resulted in a drastic increase in the GSH contents of the three tobacco cultivars (Figure 6E). The GSH contents in the drought-stressed plants were increased from 12% to 45% for N, 15% to 23% for H and 8% to 47% for Y, compared with the controls, when drought stress proceeded from 48 h to 72 h, respectively.

### 3.6. Proline and Soluble Sugar Contents in Response to Drought Stress

NC55, HongHuaDaJinYuan and Yun Yan-100 showed a significant gradual increase in the proline content of drought-treated plants in relation to its controls when drought stress passed from 24 to 72 h (Figure 5C). Its content elevated more and more as drought stress advanced. Proline content at the 24, 48 and 72 h time points of drought-stressed plants for N was increased by 19%, 117% and 191%, for H by 19%, 43% and 331% and for Y by 39%, 104% and 228%, respectively, compared with the controls. All the varieties showed a rise in the free proline content, but its rise was more prominent in H with an increment of 331% at 72 h point of drought stress. The impact of drought stress on soluble sugars (SS) at different time points (24, 48 and 72 h) for N, H and Y was found significant (Figure 5D). The percent increase in SS from control to drought-treated plants at the 24, 48 and 72 h time points for N was 20%, 31% and 10%, for H was 38%, 49% and 13%, and for Y was 35%, 25% and 25%, respectively. 

### 3.7. Leaf Anatomy in Response to Drought Stress

The leaf cross sections of all the studied varieties in this study showed two epidermal layers (upper and lower) and mesophyll parenchyma. The mesophyll parenchyma consists of palisade parenchyma having elongated cells being contacted to the upper epidermis and spongy parenchyma having small variant structure cells being contacted to lower epidermis. Leaf cross-sections of control and drought-stressed plant of the three tobacco varieties (Figure 7 and Appendix A) highlighted that they have a different responses to drought stress. HongHuaDajinYuan had no significant effect in any of the studied anatomical parameters, while N and Y had a significant effect in various anatomical parameters in response to drought stress. Total lamina of N was significantly increased in the drought-stressed plant, compared with the control while, the other two varieties had no significant variation in the total lamina. Drought stress caused an increase in upper epidermis size of N and Y, while the lower epidermis of only Y increased in size, compared with the controls. Similarly, palisade parenchyma and spongy parenchyma of only N showed an increase in its size under drought stress conditions.

### 3.8. Identification of Differentially Expressed Genes in Three Tobacco Varieties in Response to Drought Stress

In order to achieve a better understanding, a broad survey of genes associated with drought stress response in the three tobacco varieties was carried out by employing RNA-seq to analyze the transcriptome of HongHuaDaJinYuan, NC55 and Yun Yan-100 under well-watered and drought stress conditions. A total of 18 libraries were constructed and sequenced. In summary (Appendix A), 924 million raw reads were generated, 896 million total clean reads were obtained and 838 million total mapped reads were achieved onto the reference *Nicotiana tabacum* cv. K326 genome. Phred quality score Q30 ranged from 92.13% to 93.12%, with an average GC content of 42.79%. The FPKM density distribution (Appendix A) and FPKM violin diagram (Appendix A) proposed that the density of the detected genes followed a standard normal distribution.

### 3.9. Gene Ontology (GO) and KEGG Enrichment Analyses of Tobacco Transcritomes

The comparison of DEGs and their expression pattern between the libraries of the three tobacco cultivars under the same level of drought stress aid our understanding of the molecular events associated to drought stress [43]. The combine genotypic interaction analyses in response to drought stress were performed between the three tobacco varieties. As shown in Figure 8, H resulted in more number of DEGs expressed followed by Y and N in response to drought stress, respectively. Overall, 1020 DEGs were expressed between the three varieties, while 1061, 567 and 727 DEGs were expressed individually in H, N and Y during drought stress, respectively. To functionally classify those reported DEGs, the gene ontology (GO) standardized classification system for genes functions was utilized to reveal biological trends differentiating the responses of the three varieties to drought stress. The DEGs were categorized into three domains between H, N and Y comparison, including “biological processes”, “cellular components” and “molecular functions” (Appendix A). Metabolic process, oxidation-reduction process, carbohydrate metabolic process, photosynthesis etc. were the most enriched GO terms in the biological processes. Photosystem, photosynthetic membrane, thylakoid, thylakoid part, photosystem II, cell wall etc. were the most enriched gene ontology terms in the cellular component, while hydrolase activity, transferase activity, oxidoreductase activity, heme binding etc. were the most enriched GO terms in the molecular functions. To understand the functions of these DEGs, we examined the DEG-associated pathway, which were searched in the KEGG pathway database. The top 20 enriched pathways are shown in (Appendix A) in which plant hormone signal transduction, starch and sucrose metabolism, biosynthesis of secondary metabolites, photosynthesis, carbon fixation in photosynthetic organisms etc. were the most enriched pathways.

### 3.10. Plant Hormone Signal Transduction in Response to Drought Stress

Based on the GO and KEGG analyses, various DEGs were associated with the plant hormone and signal transduction pathway in response to drought stress. In total, we selected 39 DEGs that showed association with this pathway. These genes were mainly related to within abscisic acid and cytokinin signaling cascade. Four genes encoding ABA receptor PYL/PYR were downregulated under drought stress. Importantly, 13 genes were upregulated and one gene was downregulated associated to PP2C, while six transcripts expressing SnRK2 genes were both up and downregulated in response to drought stress in the three varieties. Similarly, in the cytokinin signaling cascade, three DEGs encoding cytokinin receptors (CRE) were downregulated, the inter-mediators in the cytokinin signaling, i.e., AHP, a total four genes in which three genes were upregulated and one gene was downregulated, while the eight genes encoding the response regulators, i.e., type A-ARR were downregulated in response to drought stress. A total of eight genes were selected, including four genes each associated with both hormones signaling pathways for the qRT-PCR. The comparison between the expression pattern of the selected DEGs of qRT-PCR and RNA-Seq data revealed that their expression was more or less similar to each other (Appendix A). 

### 3.11. Starch and Sucrose Metabolism in Response to Drought Stress

Our results demonstrated that drought stress induces various DEGs expression associated to the starch and sucrose metabolism on the basis of GO and KEGG analyses. In total, we selected eight genes that showed association with this pathway and were related to the soluble sugars (glucose, sucrose, fructose and maltose). Two transcripts were related to UGP (UTP-glucose-1-phosphate uridylyltransferase), two genes encoded for SUS (sucrose synthase), two genes encoded for BAM (beta amylase), one gene encoded for HXK (hexokinase) and 1 gene encoded for glgc (ADP-glucose pyrophosphorylase), which showed upregulation in response to drought stress. Five genes were selected for qRT-PCR analysis. The results of the qRT-PCR exhibited significant expression in the three varieties and the expression pattern of the selected five DEGs was more or less alike with transcriptome sequencing data (Appendix A).

### 3.12. Proline Metabolism in Response to Drought Stress

Previous studies have reported that proline play crucial role in plant growth and development under drought stress conditions in various ways like anti-oxidative defense molecule, signaling molecule etc. resulted in drought tolerance and adaptation [44,45]. Therefore, we selected various DEGs from the arginase and proline metabolism pathways. Our work demonstrated that, in total, 12 DEGs were associated with the proline metabolism in which three genes encoding P5CS (Δ1-pyrroline-5-carboxylate synthetase), four genes were related to PDH (proline dehydrogenase), two transcripts regarding to OAT (ornithine δ-aminotransferase) and three DEGs representing ARG1 (Arginase). Fours genes were selected for the qRT-PCR. The qRT-PCR analysis showed significant expression levels of the selected genes in response to drought stress in the three varieties especially in H, however the qRT-PCR expression of the selected DEGs was more or less consistent with the RNA sequence results (Appendix A).

## 4. Discussion 

Flue-cured tobacco is an important industrial crop. Water deficit stress is a major cause of crop losses that severely restricts its production. Improving drought tolerance is a challenging task and more research is needed to explore and better understand this stress. To understand this, physiological, biochemical and molecular mechanisms underlying drought tolerance would need to be exploited and studied together. Therefore, the present study was designed for better understanding the relationships between water stress and three tobacco cultivars in order to investigate the drought tolerance and adaptation mechanisms between them. Plants were subjected to well-watered and drought conditions for different duration of time. To negate the adverse effects of drought on crop production may be done by developing drought-tolerant crops. Therefore, it is important to pinpoint the drought tolerant variety and to better understand their responses to drought stress at multiple levels. In this study, HongHuaDaJinYuan performance is better and drought tolerant than Yun Yan-100 and NC55, which are the susceptible ones.

### 4.1. Growth, Chlorophyll Content and LWP of the Three Tobacco Varieties in Response to Drought Stress 

It is well known that drought is a major stress factor that reduces the crop growth. The growth inhibition in the H was ameliorated under drought stress in comparison to other two varieties. Drought stress caused 18% reduction in plant total fresh weight of H, while it caused a reduction of 23% in Y and 29% in N. Similarly, the percent reduction in plant dry weight is in order like H < Y < N (7% < 10% < 23%). In agreement with our results, [46,47] evidenced that drought stress cause reduction in plant fresh and dry weight in tobacco and tomato plants. Our results are in line with many reports that have stated that drought-tolerant varieties could maintain higher growth than drought-sensitive ones [23]. 

Chlorophyll content is the most abundant pigment in the biosphere, which indicates the photosynthetic mechanism, rate of photosynthesis and health of the plant [48]. Drought-induced chlorophyll reduction in this study is more prominent in Y and N than H, a drought tolerant variety. Chlorophyll content can be used as an efficient indicator for drought adaptive-cultivar [49]. The findings are in line with [50], as drought-tolerant varieties of *Vigna radiata* showed less reduction in chlorophyll content. 

Water potential is portrayed as a reliable parameter for measuring plant water stress response. It varied greatly, depending upon the type of the plant. Changes of leaf water potential in response to the drought stress were evaluated in various genotypes as shown in Figure 2D, the three varieties showed a clear decrease during 72 h of the drought stress. These findings are supported by [23], as drought stress decreased LWP.

### 4.2. Chlorophyll Fluorescence and Multicolor Analyses under Drought Stress

Chlorophyll fluorescence is one of the most popular techniques in plant physiology through which we can understand the fundamental mechanisms of photosynthesis and responses of plants to environmental change [51]. Chlorophyll absorbs light energy as it exists as pigment-protein complexes in PSII, PSI and within the light-harvesting complex. The fate of light energy can be (i) photo-chemistry (photosynthesis), (ii) heat dissipation and (iii) re-emitted as light (fluorescence) [52]. These three processes are in competition with each other at the same time thus chlorophyll fluorescence gives valuable information about the photochemistry and heat dissipation. Commonly used chlorophyll fluorescence parameters, such as Fv/Fm, ΦPSII_Lss and NPQ_Lss, are widely used to estimate and interpret the plant health and fitness [53]. These parameters were investigated to further explore the plant response to drought stress. 

Reduction in Fv/Fm values present characteristics of photoinhibition and the small decrease can be interpreted as photo-protection [54]. In our study, a small decline in Fv/Fm values only occurs upon 72 h of drought stress which are agreeable with [16,30]. The maximal quantum efficiency of PSII was almost unaffected until the last time point of drought stress considering generally an insensitive and might not be a good indicator for short term of drought stress. 

ΦPSII_Lss is the most important parameter that measures the proportion of light absorbed by chlorophyll associated with PSII that is used in photochemistry [52]. The drop in ΦPSII_Lss indicating the electron supply of photosynthetic carbon metabolism limited, thus, the photosynthesis was inhibited. The decrease in ΦPSII_Lss was varying in the studied genotypes. Other researchers also examined the downfall in ΦPSII_Lss values in drought-stressed plants [55], which was in harmony with our results. 

Plants curtail photo oxidative damage by keeping a balance between the absorption and utilization of light energy. Plants get rid of excessive light energy instead of adjusting light absorption and this can be achieved by dissipation of heat called non-photochemical quenching (NPQ_Lss). NPQ_Lss assessment can be presented as an indicator of damage to ATP synthase [56]. In the present research, H was the least affected among the other varieties, which indicates little damage to photosynthetic machinery. 

It is known there is a spatial heterogeneity in chlorophyll fluorescence parameters along the leaf blade of the plants as shown (Figure 4). Additionally, the explanation of spatial heterogeneity under drought is that the chlorophyll fluorescence images could display spatial variations. Different fluorescence signals of leaf tips and edges at the whole leaf might imply a greater potential restriction in photosynthesis and suppression of the protective ability in these areas [30]. 

The multicolor fluorescence imaging system is an important tool to study plant metabolism under stress conditions by monitoring signals of the blue, green and chlorophyll fluorescence [57]. In this study, as a result of drought stress, a rise in the BF, GF and RF values were witnessed in drought-treated plants of the three varieties. The increase in blue, green and red fluorescence may due to the higher content of phenolic compounds and accumulation of intermediary compounds during chlorophyll breakdown, as it is considered as an adaptation mechanism of the photosynthetic apparatus to drought stress [58]. Conversely, IrF values of all the three varieties were reduced in response to drought stress which was due to the lack of reabsorption of the emitted chlorophyll fluorescence [30]. Similarly, we witnessed an increase in the values of F690/F740 in the drought-stressed plants of the three varieties earlier on 48 h (Appendix A) than the symptoms appears in plants upon 72 h of drought stress which caused a significant reduction in chlorophyll content (Figure 2C). This ratio shows variation in the chlorophyll content and used as an indicator of the chlorophyll content [59] higher values represent lower chlorophyll content.

### 4.3. Oxidative Damage Caused by Drought Stress in the Three Tobacco Varieties

Drought stress would induce oxidative damage like lipid peroxidation, H_2_O_2_ content. Usually, the membrane lipid peroxidation in plants is detected by measuring MDA, a recognized stress biomarker [60]. In the present study, different genotypes have different levels of MDA content. The findings of the research are supported by [61], as membrane injury in term of MDA content increases in response to drought stress. Similarly, in our research, the drought-tolerant variety maintained lower accumulation of MDA content, and are in agreement with [62]. Similarly, one of the consequences of the stressor is the elevated production and accumulation of ROS such as H_2_O_2_, which causes oxidative damage to plant macromolecules and cell structures [63]. In our experiment, the overproduction of H_2_O_2_ contents was observed in each variety upon 72 h of drought stress. HongHuaDaJinYuan was the least affected cultivar with only a 14% rise in H_2_O_2_ content upon 72 h of drought stress than the other two varieties. This minimum percent decrease of H_2_O_2_ contents in H during water deficit conditions demonstrated the onset of powerful enzymatic and nonenzymatic antioxidative defense system to alleviate the oxidative damage. In our study, drought-tolerant variety maintained lower H_2_O_2_ content, as in other researches [64]. 

### 4.4. Enzymatic Antioxidant Defense System Alleviates the Drought Stress Response in the Three Tobacco Varieties

Reactive oxygen species (ROS) are the inevitable substances produced in various cellular compartments. ROS play a dual role in the plant life cycle as needed for signaling reactions, while they are toxic to plants under stressful conditions via ferroptosis (programmed pathway for cell death) [65]. It is well-known that antioxidant enzymes play a key role in protecting plants from the oxidative damage of ROS. Superoxide dismutase (SOD) is in the first line of defense against ROS, especially superoxide anion radical coupled with downstream events for detoxification of ROS. In this study, we detected significantly enhanced activity of SOD enzyme among the three varieties except for Yun Yan-100. Dong et al. [66] highlighted that drought-tolerant varieties had better SOD activity against ROS as drought stress prolonged to different time points which are in harmony with our findings. The POD activity was enhanced in all the three genotypes in response to drought stress. Apart from its protective role from oxidative damage, POD was also involved in lignification through which plants can protect themselves from drought stress [67]. Similarly, CAT is unique among other antioxidant enzymes, as it has no need of reducing equivalent and possess a high turnover rate of H_2_O_2_ to water and oxygen. As CAT and POD were both responsible for converting H_2_O_2_ into water and oxygen, the activities of the both enzymes were poorer in Yun Yan-100 than the other two varieties.

### 4.5. Non-Enzymatic Antioxidant Defense System Involved in Mitigating the Adverse Effect of Drought Stress 

Ascorbic acid (AsA), also known as vitamin C, is synthesized in the Smirnoff-Wheeler pathway, play multiple roles in plant growth and development under both stress and non-stress conditions. It is a universal non-enzymatic antioxidant that is involved in scavenging ROS and to keep its level low in plants under stress conditions. Our findings are supported by [64], as ascorbic acid pool in drought tolerant variety was more than drought sensitive varieties. Plants survival is difficult without glutathione (GSH), as it performs multiple functions in cellular defense throughout the plant life cycle. GSH content in our experiment increased with the time as drought stress prolonged in all three varieties. Awasthi et al. [68] also witnessed elevated GSH content under drought stress conditions, thus triggers adaptive responses, which is in line with our findings.

### 4.6. Proline and Soluble Sugars Improve Drought Tolerance in the Three Tobacco Varieties

Proline is a multifaceted amino acid that neither helps in plant growth but also responds to various environmental stresses. Its functions in plants are diverse in response to stress conditions. Besides an excellent osmolyte, proline plays a major role in translation (protein synthesis), keeps the redox balance in the cell, offers osmoprotection, functions as a signaling molecule (signal transduction cascade) etc. and ultimately helps in plant recovery from stress [69]. Thus, keeping the role of proline in mind, it was suggested that an increased proline concentration can contribute substantially to the plant defense system. Elevated proline contents are associated with drought tolerance and can be considered as a tool for drought tolerant genotypes [70]. In this experiment, the proline level was progressively increased in drought-stressed plants of three genotypes, as drought prolonged from 24 to 72 h compared to the controls. In particular, proline contents of H were higher than the other two varieties. The DEGs related to the metabolic pathway of proline synthesis showed higher expression in all the three varieties more specifically in H, which brought drought tolerance and adaptation in drought-stressed tobacco plants. 

Soluble sugars perform key roles in plants under stress conditions. It acts as an osmoprotectant, nutrient and signaling molecule in genes regulation that may be involved in the upregulation of growth-related genes and downregulation of stress-related genes [71]. Imposition of drought stress at different time points to the three tobacco varieties significantly increased the soluble sugar contents. Osmotic adjustment is considered to be a drought tolerant mechanism and various compatible solutes and other substances contribute to osmotic adjustment by increasing its concentration. Various studies pointed out that soluble sugar contributes to osmotic adjustment [72]. Due to its multifaceted role, it helps in drought tolerance of the studied tobacco varieties. Similarly, the DEGs related to soluble sugars metabolism in starch and sucrose metabolism pathway were highly enriched in all the three varieties and thus brought drought adaptation in tobacco plants.

### 4.7. Leaf Anatomical Modifications in the Three Tobacco Varieties under Drought Stress

Leaves are the main organ of plant and changes in the leaf physiology and anatomy are important for plants to be able to adapt to drought conditions. Measurements of the leaf structure revealed differences among the three cultivars when subjected to drought stress. Under drought stress, total lamina remains unchanged for H and Y, while it was increased for N. Thicker lamina is an adaptive strategy, and increased under drought stress [73]. Upper epidermis in N and Y and lower epidermis in Y only were increased in size in response to water deficit. An increase in the epidermal layers are also considered as an important strategy of plants to ameliorate the effects of drought stress [74] as in our observations. Mesophyll cells consist of the palisade and spongy layers which help in photosynthesis and exchange of gases, respectively. In our findings, palisade and spongy parenchyma of NC55 only showed increments in its thickness. In our study, H, a drought tolerant variety, showed no significant effect in response to drought stress in various leaf anatomical features; Hoque et al. [75] witnessed no changes in drought tolerant variety under drought stress. Observations made on the transverse sections of the leaf of three varieties, among them HongHuaDaJinYuan, had clear structures, neat and compact cells and small cell gaps under control conditions and did not show obvious changes under drought stress conditions (Figure 7), while in NC55 and Yun Yan-100, the epidermal and mesophyll cell became large and caused increased cell gaps. The cell distortion and large gaps between them were more pronounced in Yun Yan-100 (Figure 7). Our observations are in line with [76], in which drought tolerant variety have no obvious change in its anatomical features under drought stress. The leaf anatomy of the drought tolerant variety (H) was less affected than N and Y, which might indicate that the role of UGP genes bring cell wall modifications in response to drought stress as in cotton [77]. Their expression levels were higher in H than in the other two varieties. The tolerant varieties must be those that perform well and adaptive features only do not indicate higher tolerance under drought stress but the cultivar also must not deviate from its normal growth [75].

### 4.8. Analysis of Plant Hormone Signal Transduction in Response to Drought Stress

As plants are sessile organisms, they often face a combination of biotic and abiotic stresses. It is noteworthy how crosstalk occurs in plants and switches the responses between the various pathways [78]. Plant hormones may be the key regulator in term of the responses to the detrimental effect of various forms of stresses and coordination between the different hormone signaling could be flexible in term of type and intensity of stress and type of plant under the stress [79]. Abscisic acid (ABA) regulates key processes throughout the plant life cycle under both normal and adverse environmental conditions that lead to the expression of genes related to the specific condition by initiating the cell signaling pathways [80]. PYL, PP2C and SnRK2 are the key components of the ABA signaling pathway as PYL are the receptors; PP2C acts as a negative regulator, while SnRK2 acts as a positive regulator in ABA signaling [81]. Our findings showed that all the PYL gene were downregulated in response to drought stress, which are in line with [82]. PP2C is the intermediary component in ABA signaling pathway, as in our study, overall, the PP2C genes expression were upregulated under drought stress conditions, which is in accordance with the findings of [83], as PP2C genes were upregulated in response to drought stress which play a role in plant drought tolerance. The SnRK2 are the plants specific kinases and key regulators of the ABA signaling pathway under abiotic stress conditions [84]. In our findings, the SnRK2 genes showed variation in their expression, as some genes were upregulated, while some genes were downregulated. This is in line with [85], which indicated its conserved as well as diverse roles in response to drought stress. The upregulation of the SnRK2 genes signifies its role in stomata movements as its physical interaction with K^+^ channels (KAT1), anion channels (SLAC1) and NADPH oxidases [86] under drought stress.

Cytokinins are important ubiquitous plant hormones that regulate plant growth and development in a sustainable way under unfavorable environment conditions. In our study, the cytokinin receptors CRE/AHK4, AHK2, AHK3 were downregulated in response to drought stress. The various cytokinin receptors genetic analysis exhibited drought tolerance in *Arabidopsis* suggesting its role in response to various abiotic stresses conditions [87]. The histidine containing phospho-transmitter (AHP) is a key component in the cytokinin signaling pathway; genes related to AHPs were upregulated in our findings which are in line with [88], as was expressed in *Brassica rapa,* which plays a pivotal role in drought stress. The hormonal stress adaptation is the result of various antagonistic and synergistic interactions between various hormones [89]. In our results, both the AHP and SnRK2 genes were expressed in drought stress. This shows the crosstalk of cytokinin and abscisic acid as both hormones interact antagonistically in mediating drought stress [19]. Figure 9A exhibited the proposed pathway of the selected DEGs, which results in bringing drought tolerance. Figure 9B explains that the expression pattern of the DEGs which were involved in plant hormone signaling pathway in response to drought stress. Most of the DEGs related to PP2C and SnRK2 exhibited higher upregulation expression and the DEGs related to PYL and A-ARR showed higher downregulation expression in the drought tolerant variety (H) than the other two varieties suggesting its decisive role in the drought stress tolerance. 

### 4.9. Analysis of Starch and Sucrose Metabolism in Response to Drought Stress

Plants adapt various mechanisms to negate the effects of the adverse environmental conditions, as carbohydrates and sugar metabolism play pivotal role in stress responses and are considered as key determinant in plant fitness and bringing tolerance to various abiotic stresses [90]. Soluble sugars are highly sensitive to environmental stress and some sugars bring plant tolerance especially to drought stress. Sugars are the primary products of plants, which are used in all metabolic pathways and need to be phosphorylated, and to be metabolized; hexokinases (HXK) phosphorylates the hexose. Hexokinases play multiple roles in ever-changing environmental conditions, as it brings coordination between photosynthesis and transpiration rate and also helps in controlling stomatal movements during water deprivation [91]. Our results showed that genes encoding hexokinases were upregulated under drought stress. Maltose, a soluble sugar, the product of beta amylase (BAM), plays a protective role and brings acclimation to temperature stress [92]. The expression of BAM encoding transcripts were upregulated in response to drought stress. Sucrose synthase (SUS) is a key enzyme in sucrose metabolism, as sucrose is important for cell growth. In our results, SUS related gene expression was upregulated under drought stress, which is supported by [93], as water cessation cause higher expression of SUS in wheat. UDP-glucose is an essential nucleotide sugar which is necessary for sucrose and cell wall bio-synthesis and UGP is a key enzyme in this pathway. In our findings, the UGP genes expression was higher and upregulated in response to drought stress in drought tolerant cultivar. AGPase is the first rate limiting enzyme in starch bio-synthesis and induced by various abiotic stresses and enhances stress tolerance. In our results, the genes encoding AGPase (*glgc*) were highly upregulated in the drought tolerant variety as compared to the other two varieties as its expression was also higher in maize tolerant variety [94]. Figure 10A showed the proposed pathway of the genes involved in the regulation of starch and sucrose metabolism which helps in the amelioration of drought stress effects. The starch and sucrose metabolism pathway comprised of glgc, UGP, SUS, BAM and HXK enzymes related genes were all upregulated in the three varieties under drought stress, but their expression levels were high in H following N and Y (Figure 10B), indicating that it might be involved in drought tolerance as in cucumber under drought stress [16]. 

### 4.10. Analysis of Proline Metabolism in Response to Drought Stress

Among the various abiotic stresses, drought is the most destructive one which puts the plants in jeopardy. Plants evolve various defense mechanisms to survive under the specific stress condition, in which accumulation of compatible solutes (polyamines, glycine betain, soluble sugars, proline etc.) occur and bring osmoregulation, which is an adaptive mechanism in drought tolerance [95]. Among the various compatible solutes, proline is an excellent osmolyte that performs various functions, such as antioxidative defense molecule, signaling molecule and buffering of cellular redox potential under stress conditions [69,96]. Proline metabolism occur in plants via glutamate-derived and ornithine-derived metabolic pathways; Δ^1^-pyrroline-5-carboxylate synthetase (P5CS) converts glutamate into glutamate-γ-semialdehyde (GSA), which is an intermediary compound in proline synthesis, while ornithine-δ-aminotransferase (OAT) converts ornithine into GSA (Figure 11A) [97]. In our findings, genes related to proline biosynthesis in both glutamate and ornithine-related metabolic pathways were upregulated in response to drought stress which are in line with for P5CS related genes [98], for OAT [99] and for proline dehydrogenases (PDH) [100]. Our findings are showing that proline metabolism and turnover occur, instead of just proline accumulation, suggesting that growth is maintained under drought stress [101]. Similarly, arginase genes (ARG) convert arginine into ornithine, which shows that proline metabolism was induced by drought stress. Various studies pointed out that manipulation in ARG genes showed tolerance to drought stress [102]. The DEGs related to this pathway which comprised of the key enzymes (P5CS, PDH, OAT and ARG1) involved in proline metabolism were upregulated in all the three varieties. Thus, in this study, the genes related to the proline metabolic pathway were highly expressed in H, a drought tolerant variety, compared to the drought sensitive varieties (Figure 11B).

## 5. Conclusions 

In summary, this study provides a comprehensive analysis at a physiological, biochemical, anatomical and transcriptome level in response to drought stress. Three tobacco varieties were subjected to drought stress, which showed a diverse drought responses. HongHuaDaJinYuan, a drought tolerant variety, showed a better performance than the other two varieties at physiological, biochemical, anatomical and transcriptome level. The better performance of HongHuaDaJinYuan can be seen as lower reduction in plant biomass and chlorophyll content with a powerful antioxidant defense system to alleviate oxidative damage under drought stress. Similarly, the expression of genes related to the hormone signal transduction, starch and sucrose metabolism, and arginine and proline metabolism, were more pronounced in H compared to the other two varieties under drought stress. Overall, the data presented here will be useful to further understanding the drought tolerance and adaptation mechanisms in plants.

## Figures and Tables

**Figure 1 genes-10-01041-f001:**
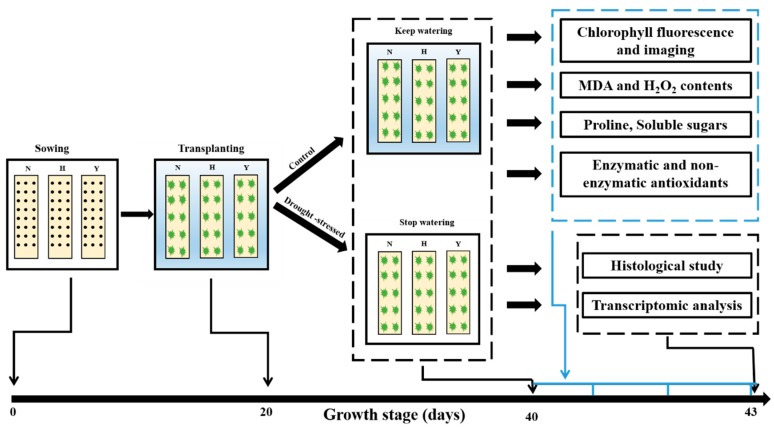
Protocol of drought stress treatment of the three tobacco varieties, i.e., N, H and Y representing NC55, HongHuaDaJinYuan and Yun Yan-100, respectively.

**Figure 2 genes-10-01041-f002:**
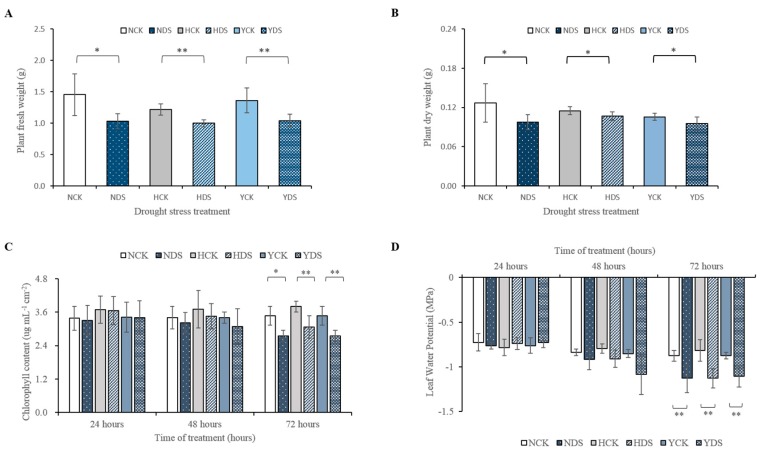
Effect of drought stress on plant fresh weight (**A**), plant dry weight (**B**), chlorophyll content (**C**) and leaf water potential (LWP) (**D**). Bars are the means of ± SD of three replicates. * LSD, *p* ˂ 0.05, ** LSD, *p* ˂ 0.01. NCK stands for NC55 under normal conditions; NDS stands for NC55 under drought stress conditions; HCK stands for HongHuaDaJinYuan under normal conditions; HDS stands for HongHuaDaJinYuan under drought stress conditions; YCK stands for Yun Yan-100 under normal conditions; YDS stands for Yun Yan-100 under drought stress conditions.

**Figure 3 genes-10-01041-f003:**
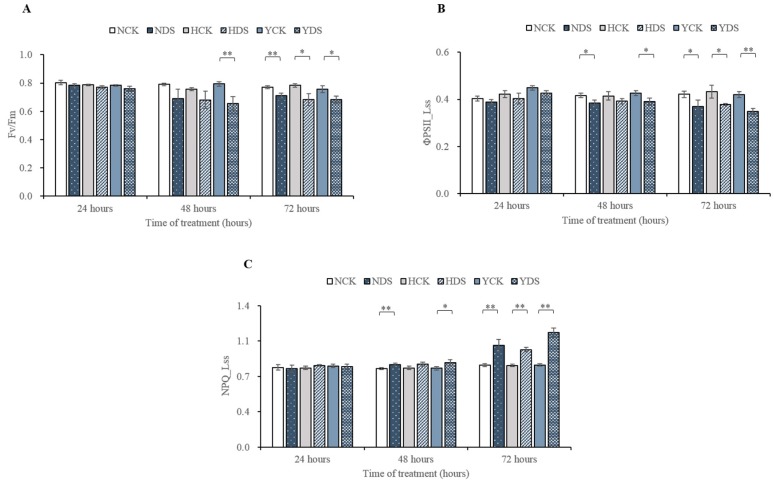
Effect of drought stress on Fv/Fm (**A**), ΦPSII_Lss (**B**) and NPQ_Lss (**C**) of the three tobacco varieties over time. Bars are the means of ± SD. * LSD, *p* ˂ 0.05, ** LSD, *p* ˂ 0.01. NCK stands for NC55 under normal conditions; NDS stands for NC55 under drought stress conditions; HCK stands for HongHuaDaJinYuan under normal conditions; HDS stands for HongHuaDaJinYuan under drought stress conditions; YCK stands for Yun Yan-100 under normal conditions; YDS stands for Yun Yan-100 under drought stress conditions.

**Figure 4 genes-10-01041-f004:**
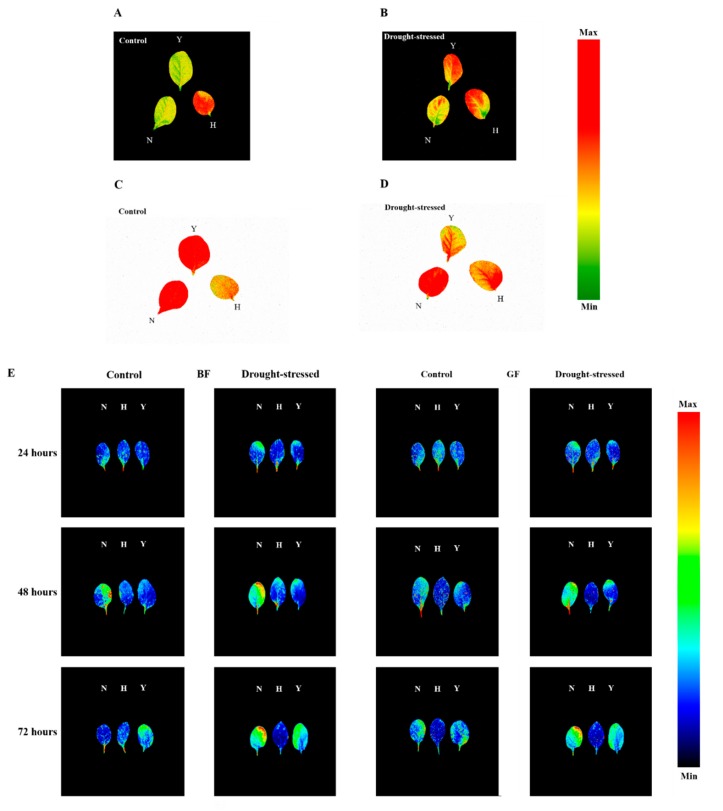
Representative chlorophyll fluorescence images of non-photochemical quenching (**A**,**B**), and PSII quantum yield (**C**,**D**) during light adaptation and multicolor fluorescence images of blue fluorescence (**BF**) and green fluorescence (**GF**) (**E**) of the three tobacco varieties under control and drought stress treatment respectively. N represents NC55; H represents HongHuaDaJinYuan; Y represents Yun Yan-100.

**Figure 5 genes-10-01041-f005:**
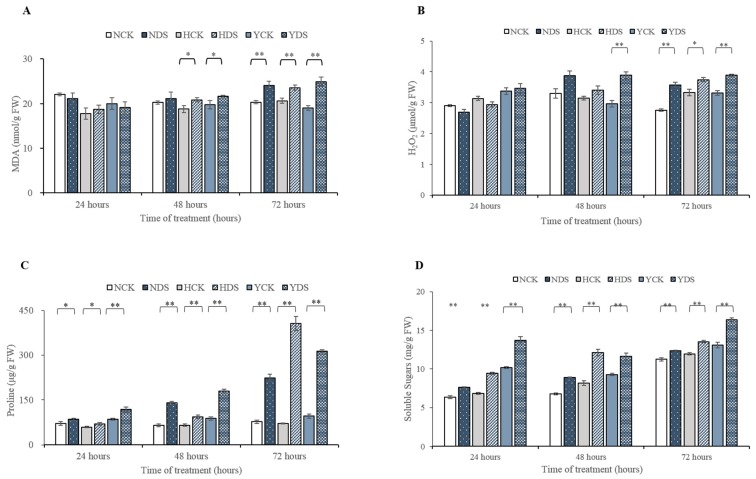
The contents of MDA (**A**), H_2_O_2_ (**B**), Proline (**C**) and Soluble Sugars (SS) (**D**). Plants were subjected to drought stress for different time course along with their respective controls. Data are the means of ± SD (three replicates, * LSD, *p* ˂ 0.05, ** LSD, *p* ˂ 0.01 by LSD test). NCK stands for NC55 under normal conditions; NDS stands for NC55 under drought stress conditions; HCK stands for HongHuaDaJinYuan under normal conditions; HDS stands for HongHuaDaJinYuan under drought stress conditions; YCK stands for Yun Yan-100 under normal conditions; YDS stands for Yun Yan-100 under drought stress conditions.

**Figure 6 genes-10-01041-f006:**
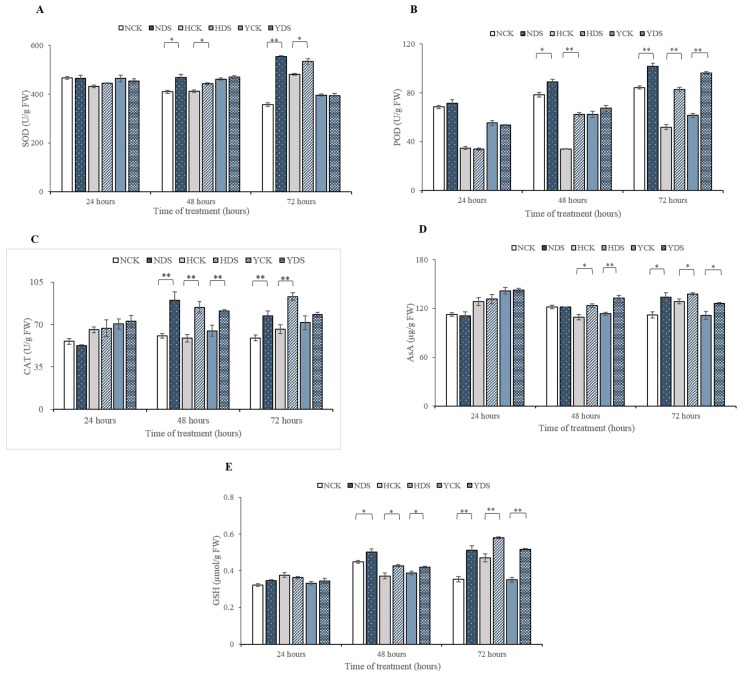
Drought stress induced changes in the activity of three main antioxidant enzymes, i.e., SOD (**A**), POD (**B**) and CAT (**C**), and two non-enzymatic substances, i.e., AsA (**D**) and GSH (**E**), of the three tobacco varieties. Means are represented with error bars ± SD of three replicates, * *p* ˂ 0.05, ** *p* ˂ 0.01 by LSD test. NCK stands for NC55 under normal conditions; NDS stands for NC55 under drought stress conditions; HCK stands for HongHuaDaJinYuan under normal conditions; HDS stands for HongHuaDaJinYuan under drought stress conditions; YCK stands for Yun Yan-100 under normal conditions; YDS stands for Yun Yan-100 under drought stress conditions.

**Figure 7 genes-10-01041-f007:**
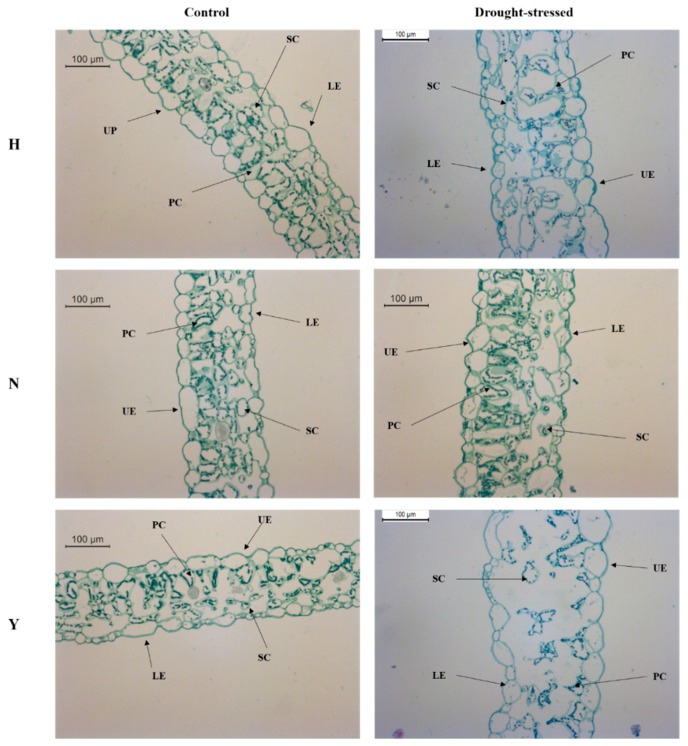
Representative leaf anatomical images of the three tobacco varieties under control and drought stress treatment. UE: upper epidermis, LE: lower epidermis, SC: spongy cells and PC: palisade cells. H represents HongHuaDaJinYuan; N represents NC55; Y represents Yun Yan-100.

**Figure 8 genes-10-01041-f008:**
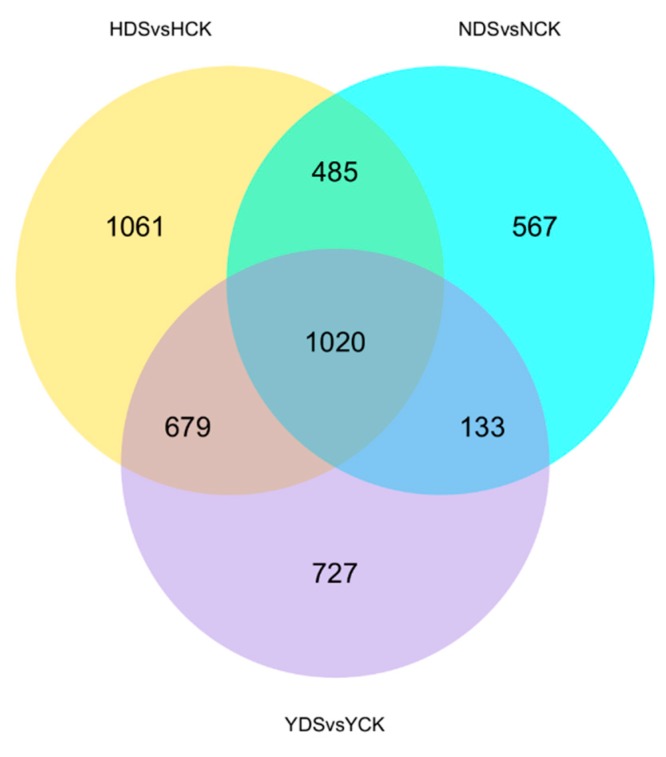
The dynamic progression of the tobacco transcriptome was effected by genotype and drought stress. A Venn diagram of the numbers of differentially expressed genes (DEGs) in comparison between H, N and Y. H represents HongHuaDaJinYuan; N represents NC55; Y represents Yun Yan-100. HCK stands for HongHuaDaJinYuan under normal conditions; HDS stands for HongHuaDaJinYuan under drought stress conditions; YCK stands for Yun Yan-100 under normal conditions; YDS stands for Yun Yan-100 under drought stress conditions; NCK stands for NC55 under normal conditions; NDS stands for NC55 under drought stress conditions.

**Figure 9 genes-10-01041-f009:**
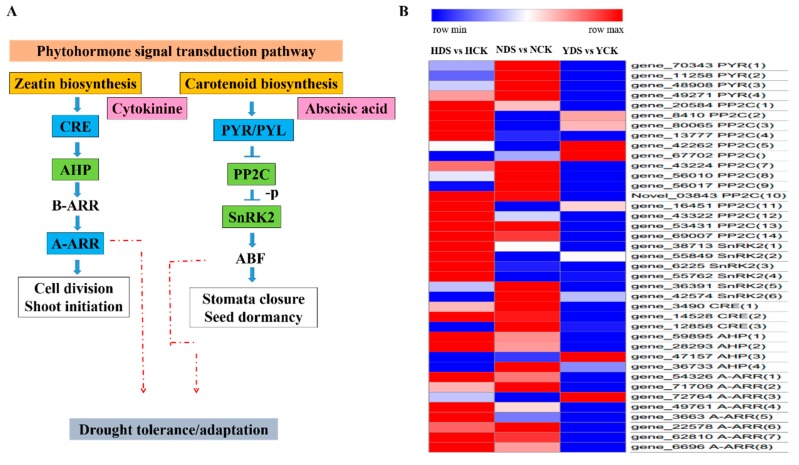
(**A**) DEGs involved in phytohormone signal transduction pathway. The proposed pathways show phytohormones biosynthesis. This figure shows only the genes that have been associated with the pathway from transcriptome analysis. Dark pink boxes show plant hormones, light blue boxes show the downregulation of genes in response to drought, light green boxes show both up and downregulated genes, red arrows show the involvement in the drought tolerance. -p indicates de-phosphorylation process. CRE (Cytokinin receptors), AHP (authentic histidine-containing phosphotransmitter), ARR (authentic response regulator), PYR (pyrabactin resistance-ABA receptor), PP2C (Protein phosphatase 2C), SnRK2 (Serine/threonine-protein kinase). (**B**) Heat map of selected DEGs showing their expression behavior corresponding to **“A”**. Red and blue indicate higher and lower expression values, respectively. NCK stands for NC55 under normal conditions; NDS stands for NC55 under drought stress conditions; HCK stands for HongHuaDaJinYuan under normal conditions; HDS stands for HongHuaDaJinYuan under drought stress conditions; YCK stands for Yun Yan-100 under normal conditions; YDS stands for Yun Yan-100 under drought stress conditions.

**Figure 10 genes-10-01041-f010:**
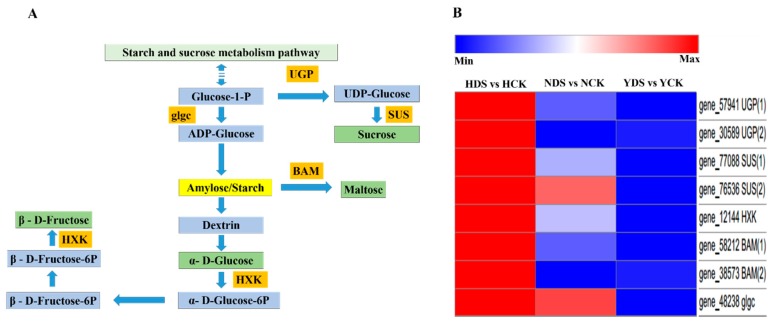
(**A**) DEGs involved in starch and sucrose metabolism pathway. The proposed pathway shows starch metabolism. This figure shows only the genes that have been associated with the pathway from transcriptome analysis. Yellow box shows Starch. Green boxes show soluble sugars (sucrose, glucose, fructose and maltose), blue boxes show the downregulatory components in the pathway. Orange boxes show upregulated enzymes in response to drought stress. SUS (Sucrose synthase), BAM (beta amylase), AMY (alpha amylase), HXK (hexokinase), glgc (AGPase-ADP-glucose pyrophosphorylase), UGP (UTP-glucose-1-phosphate uridylyltransferase). (**B**) Heat map of selected DEGs showing their expression behavior corresponding to **“A”**. Red and blue indicate higher and lower expression values, respectively. NCK stands for NC55 under normal conditions; NDS stands for NC55 under drought stress conditions; HCK stands for HongHuaDaJinYuan under normal conditions; HDS stands for HongHuaDaJinYuan under drought stress conditions; YCK stands for Yun Yan-100 under normal conditions; YDS stands for Yun Yan-100 under drought stress conditions.

**Figure 11 genes-10-01041-f011:**
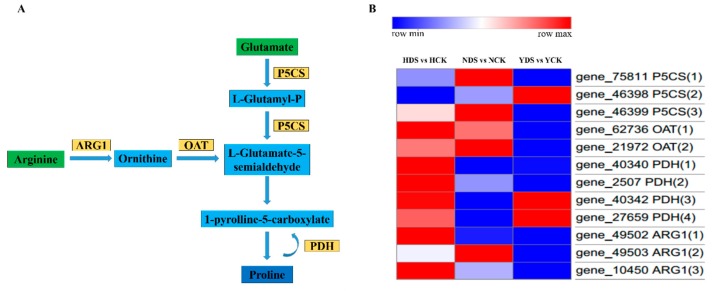
(**A**) DEGs involved in Arginine and Proline metabolism pathway. The proposed pathway shows proline metabolism. This figure shows only the genes that have been associated with the pathway from transcriptome analysis. Green boxes show the pathways of proline biosynthesis. Blue box shows Proline. Light blue boxes show the regulatory components in the proline metabolism pathway. Orange boxes show upregulated enzymes in response to drought stress. ARG1 (Arginase), OAT (ornithine δ-aminotransferase), P5CS (Δ1-pyrroline-5-carboxylate synthetase), PDH (proline dehydrogenase). (**B**) Heat map of selected DEGs showing their expression behavior corresponding to **“A”**. Red and blue indicate higher and lower expression values, respectively. NCK stands for NC55 under normal conditions; NDS stands for NC55 under drought stress conditions; HCK stands for HongHuaDaJinYuan under normal conditions; HDS stands for HongHuaDaJinYuan under drought stress conditions; YCK stands for Yun Yan-100 under normal conditions; YDS stands for Yun Yan-100 under drought stress conditions.

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
