# Peer review of "Transcriptome Profiling, Biochemical and Physiological Analyses Provide New Insights towards Drought Tolerance in Nicotiana tabacum L."

_genes, 2019, doi:10.3390/genes10121041_

Round 1

Reviewer 1 Report

The work presented by Rayyan Khan and colleagues provide a wide characterization of physiological, anatomical, biochemical and transcriptomic changes occurring in Tobacco during drought stress. The work can be of potential interest because of the topic investigated. To avoid that the manuscript results in a sterile descriptive exercise, a depth revision of the manuscript is required to provide clear insights on the relevant results for the topic.  

Major comments:

The main issue of the presented work is that the specific objective of the paper is not fully clear. Is it: “To unravel the various mechanisms underpinning tobacco (Nicotiana tabacum L.) drought stress tolerance through a comprehensive physiological, anatomical, biochemical and transcriptome analysis of three tobacco cultivars” as indicated in the abstract Or “to investigate the difference of drought tolerance between them [three tobacco cultivars].” As indicated in the discussion

Answering one or the other objective implies different result analysis and interpretation. As for example:

To provide a comprehensive analysis on the various mechanisms underpinning tobacco drought stress tolerance, one would focus on common pathways modulated in the three cultivars by drought stress To investigate the differences and identify pathway of drought-stress resistance, one would focus more on pathways more specific (or even unique) for the stress-resistant cultivar(s).

The results are presented without a clear point of view, therefore it is not clear what molecular mechanisms are actually linked to resistance and what to general drought-stress response. A more clear cut must be given, orienting the analysis in one or the other direction.

Introduction misses to explain in depth what is the current knowledge on drought-stress mechanisms in Tobacco, what knowledge is missing and what the presented work is aiming to add to the state of art.

The manuscript is extremely long and this doesn’t allow the reader to have a comprehensive view and understanding of results, including their consistency. Text should be consistently reduced providing a more concise description and discussion of results.

All sections are filled with didactic explanations that are not necessary, while they frequently miss a clear interpretation of results and take-home messages. Not-useful sentences include for example:

Intro: “To explore the stress resistance mechanisms in plants, the transcriptome sequencing renders the most direct and efficient way due to the advancements in high-throughput sequencing technology with least cost. The next generation sequencing (NGS) are efficient method to obtain a depth of sequencing that is sufficient to cover the transcriptome of an organism many fold and allow quantification of the detected transcripts in plants. “ it is established in literature that RNAseq is a suitable technique for transcriptome analysis, no need to point this out in such a manuscript.

Results: “Normalization is an important  consideration for the reliability of RNA-seq data analysis which has basic importance for discovering biologically important changes and leading to the more accurate calculation of genes expression [36]”. Normalization is mandatory indeed, but it is method, no need to be presented as a result.

Reusults. “Chlorophyll contents are the most abundant pigments in the biosphere which actively  participate in photosynthesis by harvesting light energy and driving electron transport.” inappropriate sentence for result section, I’m sure readers will know the importance of Chlorophyll in plants.

Many other examples of these type are present in the manuscript.

Conclusions on results are not always fully correct, therefore the text should be extensively revised to avoid result over-interpretation. Examples include:

Line 444. “These results indicated high quality and reasonable reproducibility of sequence data.” Reproducibility of results cannot be assessed on the basis of this analysis (violin plot of FPKM), but based on comparison of replicates. In any case such analysis is out of the scope of the manuscript.

Line 535. “All the selected genes were significantly deferentially expressed, and the expression profile were consistent with transcriptome sequencing data.” PP2C, CRE, ARR and AHP expression are not fully consistent with RNAseq results. Similar considerations for other replication analysis.

Figure 15, 17, 19 and related discussion paragraph: it is not clear what genes have been identified as up or down-regulated. Not even if the pathway is overall induced or repressed upon drought-stress or if it is linked to drought-stress resistance.

On which samples the validation of RNAseq data by RTqPCR has been performed? It is not clear if additional samples have been analysed other than those used for RNAseq. If the same samples have been used, the value of such analysis is very limited and must be moved to supplementary. In addition primer design strategy in relation to RNAseq data should be outlined.

It is not clear why the authors chose to focus on the validation and in depth description of some pathways while excluding others. This must be justified, also in light of point 1 (aim).

It is not clear on which set of differentially expressed genes Go-term and pathway-enrichment analysis have been performed. This is again crucial, and interpretation of results is dependent both on the selection of specific vs common genes and in light of point 1 (aim). If the analysis was performed on all DEG, it is conceptually not correct instead.

Showing three Go-term and pathway-enrichment analysis is unnecessarily long, it makes difficult to pin-point differences and consistencies between different analysis. Results must be summarized in a unique table/figure, and subsequently interpreted on the basis of point 1 (aim).

A more clear description and discussion of transcriptomic changes in relation to physiological anatomical, biochemical changes must be provided.

A study with a very similar experimental setup and aims has been recently published (Front. Plant Sci., 17 May 2017 | https://doi.org/10.3389/fpls.2017.00827). However a clear comparison of results is missing as well as the added value of the current study as compared to this other work.

Minor:

Figure legends must always carefully report all acronyms used in the related figure. Figure 16 and 18 are missing M&M: please specify type of sequencing, kit for RNAseq library prep (if custom protocol provide more extensive description), Retrotranscription kit and conditions. Venn diagrams must be all included in a single figure or table.

Reviewer 2 Report

Dear Authors,

In this article, Khan and colleagues perform a comprehensive time-course investigation into the impact of drought stress on three tobacco cultivars. They performed exhaustive analyses at the anatomical, physiological, biochemical and gene expression levels – the results/raw data of which are very valuable resources to the academic community. Based on a combination of these approaches, they concluded that the cultivar HongHuaDaJinYuan (H) is the most drought-tolerant. This study will be of great interest to both fundamental plant biologists and applied crop scientists alike, as it provides insights on both a mechanistic level and also at a translational perspective.

To further improve the manuscript, there are some issues that the authors need to address as outlined below:

Because it is referring to the plural form, the title of the manuscript could be revised to “Transcriptome profiling, biochemical and physiological analyses…” instead of “…analysis…” Although the cultivar H had the most drought-tolerant parameters based on several assays, the differences with cultivars NC55 (N) and Yun Yan-100 (Y) were not hugely dramatic in some cases (albeit statistically significant). In order to show evidence of the robustness, consistency and reproducibility of the observations, the authors need to state how many times the experiment outlined in Figure 1 (Section 2.1) was repeated independently. Please provide more details on the RNA extraction procedure (Section 2.10). Which method (Trizol, phenol, etc.) and/or kits were used? Because the main thesis of the article is that cultivar H is the most drought-tolerant, I would recommend showing representative time-course non-fluorescence photos of how the plants looked throughout the drought treatment when compared to control plants. In Section 3.4, the Conclusion “Overall, the performance of HongHuaDaJinYuan was better than NC55 and Yun Yan-100 because the activity of all the studied antioxidant enzymes was enhanced to drought stress as the stress conditions prolonged from 24 hours to 72 hours” is not completely valid as cultivar N had better SOD performance (see Figure 8A). The concluding sentence need to be qualified and clarified to include this observation. Also, in Section 3.4, there needs to be a comment on the already high basal POD levels in cultivar N compared to H and Y. Even though the percentage increase in POD was lower in N, it remained to have the highest absolute levels. This would suggest that, in terms of POD activity levels, cultivar N had the highest POD antioxidant activity instead of H (as stated by the authors). In the transcriptome analyses of the paper (Sections 3.8-3.9), it would have been preferable to perform a combined genotype X drought interaction analysis. The results would have been more informative as it would have answered (at the molecular level) what makes cultivar H the most drought-tolerant among the three cultivars. The current three pairwise comparisons (H vs. N; N vs. Y, H vs Y) dilutes the main thesis of the paper. Although cultivar H had more genes with the highest differential expression, there were instances when the other cultivars had higher gene expression changes. This is not reflected in the conclusion in Section 3.10, as it makes a broad claim for all genes. I would recommend that the authors qualify their statement.

Specific comments on Figures and Tables:

All Figure captions: Please make sure to explicitly specify in all the Figure Captions what all the acronyms are. What do N, H and Y stand for? What do NCK, NDS, etc. stand for? Table S2: Please indicate what the acronyms H, N, Y, CK and DS in the Table caption. Also, please indicate the standard deviations – only the means are currently shown.

Specific comments on the text:

Line 23, Page 1: Correct to “…least affected” Line 24-25, Page 1: Please restate and clarify the phrase “… with as less percentage increase in ROS.” Line 38, Page 1: Please restate to “Drought is one of the most eminent environmental stresses…” Line 62, Page 2: Correct to “… others show adaptive processes…” Line 102, Page 3: Correct to “… and drought-treated plants (DS)…” Line 116, Page 3: Correct to “… dark- and light-adapted leaves…” Line 342, Page 11: Correct to “… in terms of SOD activity was poor as there was no significant difference…” Line 440, Page 14: Correct to “… with an average GC content of…” Lines 593-597, Page 21: These statements need to have references. Line 851, Page 27: Correct to “… presented here will be useful…”

Round 2

Reviewer 2 Report

Dear Authors,

The following issues have been addressed, which further improved the manuscript:

The title of the paper has been revised slightly to address correct syntax. The authors have stated how many times the experiment in Figure 1 (Section 2.1) was repeated independently and they defended their rationale appropriately. More details on the RNA extraction procedure (Section 2.10) have been provided. The authors have added representative photos of the drought-induced phenotype of all three cultivars. In Section 3.4, the Conclusion has been clarified to state that cultivar N had better SOD performance and the highest absolute levels of POD activity. Detailed descriptions and acronym explanations have been provided in the figure and table labels.

The following minor issues still need to be properly addressed:

The transcriptome analyses (Sections 3.8-3.9) have been redone with a combined genotype X drought interaction analysis, instead of the previous pairwise comparisons (H vs. N; N vs. Y, H vs Y). However, the GO and KEGG pathway enrichment still need to be performed with a combined analyses, in order to provide molecular insights into why cultivar H is the most drought-tolerant of the three. In Section 3.10, the authors state, “The comparison between the expression pattern of the selected DEGs of qRT-PCR and RNA-Seq data revealed that their expression was more or less similar to each other (Figure S10). The results of the qRT-PCR showed that its expression were more pronounced in HongHuaDaJinYuan as compared to the other two varieties in response to drought 636 stress.” – the second sentence is not consistent with the newly added first sentence (which accurately reflects the data presented). This needs to be corrected by removing the second sentence.
